# Influence of the Land Use Type on the Wild Plant Diversity

**DOI:** 10.3390/plants9050602

**Published:** 2020-05-08

**Authors:** Ina Aneva, Petar Zhelev, Simeon Lukanov, Mariya Peneva, Kiril Vassilev, Valtcho D. Zheljazkov

**Affiliations:** 1Institute of Biodiversity and Ecosystem Research, Bulgarian Academy of Sciences, 1113 Sofia, Bulgaria; Simeon_Lukanov@abv.bg (S.L.); Kiril5914@abv.bg (K.V.); 2Department of Dendrology, University of Forestry, 1797 Sofia, Bulgaria; petar.zhelev@ltu.bg; 3Department Economics of Natural Resources, University of National and World Economy, 8th December Blvd., 1700 Sofia, Bulgaria; Peneva_mm@yahoo.co.uk; 4Department of Crop and Soil Science, College of Agricultural Sciences, Oregon State University, Corvallis, OR 97331, USA; valtcho.jeliazkov@oregonstate.edu

**Keywords:** sustainable agriculture, biodiversity, flora, bioclimatic variables

## Abstract

Studies on the impact of agricultural practices on plant diversity provide important information for policy makers and the conservation of the environment. The aim of the present work was to evaluate wild plant diversity across the agroecosystems in two contrasting regions of Bulgaria; Pazardzhik-Plovdiv (representing agroecosystems in the lowlands) and Western Stara Planina (the Balkan Mountains, representing agroecosystems in the foothills of the mountains). This study conducted a two-year assessment of plant diversity in different types of agricultural and forest ecosystems, representing more than 30 land use types. Plant diversity, measured by species number, was affected by the land use type only in Pazardzhik-Plovdiv region. More pronounced was the effect of the groups of land use types on the diversity, measured by the mean species number per scoring plot. Climatic conditions, measured by 19 bioclimatic variables, were the most important factor affecting plant species diversity. Six bioclimatic variables had a significant effect on the plant diversity, and the effect was more pronounced when the analysis considered pooled data of the two regions. The highest plant diversity was found on grazing land with sparse tree cover, while the lowest one was in the land use types representing annual crops or fallow. The study also established a database on weed species, relevant to agriculture. A number of common weeds were found in the Pazardzhik-Plovdiv region, while the most frequent species in the Western Stara Planina region were indigenous ones. Overall, the natural flora of Western Stara Planina was more conserved; eleven orchid species with conservation significance were found in the pastures and meadows in that region. The present study is the first attempt in Bulgaria to characterize the plant diversity across diverse agroecosystems representing many different land use types and environmental conditions. The results can contribute to nature conservation, biodiversity, and the sustainable use of plant resources.

## 1. Introduction

Agricultural activity is one of the main factors causing a decrease in biodiversity globally [1]. Humans have been converting natural ecosystems into agricultural land for millennia and this process continues in some areas across the globe. This conversion generally results in the loss of soil carbon and in reduced plant, animal, and microbial biodiversity [2]. While practicing a small-scale traditional agriculture during the pre-industrial period still allowed co-existence of natural biodiversity with the agricultural ecosystems, the situation changed dramatically with the intensification of agriculture. Land consolidation and enlargement of agricultural fields lead to the removal of field borders, hedges and other microhabitats, and the use of heavy machinery, pesticide applications and intensive tillage also negatively affected the plant biodiversity. As a result, the agricultural landscapes today differ considerably from the natural ones [3]. 

Generally, the term biodiversity is applied to agriculture in two aspects; (1) diversity of crops and crop cultivars, and (2) diversity of wild species growing within the fields and at the field borders (edges). While the first aspect of diversity is related to increasing crop yields and resilience of managed ecosystems, the second one focuses more on the conservation of biodiversity in agricultural ecosystems as a whole. Sometimes there could be a conflict between the objectives of the two aspects, because crop quality and yield increase may require more frequent pesticide application and tillage to reduce the weed seed bank, but that may also reduce plant biodiversity. These issues have always been a concern for the sustainable agriculture proponents [4]. In attempt to resolve the controversies, the European Commission introduced agri-environmental measures aimed generally at integrating the environmental concerns into agricultural policy and, in particular, at the conservation of biodiversity on agricultural land [5]. Among the particular tasks of agri-environmental schemes are the maintenance and conservation of so-called Green and Blue Infrastructure (GBI), consisting of different natural microhabitats within the agricultural land such as field margins, wetlands, small streams, shelterbelts, and woodland spots [6]. An example is the introduction of the sub measure “Pastoralism” in Bulgaria, as an important tool for conservation of grassland habitats [7]. According to Duelli and Obrist [8], flowering plants are among the four groups of living organisms (together with insect groups Heteroptera, Symphyta and aculeate Hymenoptera) presenting the best choice for biodiversity evaluation in cultivated areas. The plant diversity distribution across the habitats depends on many factors, and on agricultural land it is strongly affected by the land use. Titeux et al. [9] emphasized that biodiversity scenarios mostly consider the future impacts of climate change, while the changes in land use and land cover remain largely neglected. This is an important omission, since the direct habitat destruction through land use is among the most important and immediate threats to biodiversity.

Generally, there are insufficient scientific data on the impact of land use style on plant species biodiversity in the Balkans. Therefore, the objective of the present study was to assess the effect of land use on the flowering plant diversity in two regions of Bulgaria: Plovdiv-Pazardzhik region and Western Stara Planina. The two regions differ in traditional and current agriculture systems and land use and present a good opportunity for biodiversity assessment of two contrasting agricultural ecosystems [10,11,12]. 

## 2. Results and Discussion 

Studying the effect of environmental factors on the distribution of biodiversity requires ranking of these factors according to the magnitude of their effect [13]. Doubtless, the climatic factors are the major determinants of species distribution patterns [14]. Therefore, the selection of the most appropriate bioclimatic variables was a necessary prerequisite for the analysis. 

The first three components of the bioclimatic variables explained 94% of the total variance (Table 1). Principal component 1 (PC1) explained 54% of the total variation. Of the temperature-related variables, PC1 was positively correlated with the mean diurnal range (0.299), isothermality (0.291), and mean temperature of the driest quarter (0.176), and negatively correlated with the mean temperature of the wettest quarter (−0.272). PC1 was also strongly and negatively correlated with the variables related to humidity (seven correlation coefficients ranging from −0.297 to −0.308; Table 1). 

Principal component 2 (PC2) explained an additional 27% of the variation and was correlated most strongly with mean annual temperature (0.441), maximum temperatures of the warmest and minimum temperatures of the coldest month (0.349 and 0.404, respectively), and mean temperatures of the quarters (Table 1). Principal component (PC3) was correlated strongly and positively with temperature seasonality (0.592), temperature extremes and temperature annual range, and negatively with precipitation seasonality (−0.494), and mean temperature of the driest quarter (−0.303) (Table 1).

Selection of appropriate variables is of key importance for the modeling of plant species distribution. There are some shortcomings related to the most commonly used variables, most of which are based on data averaged across seasons and years [13]. For example, seasonal or annual rainfall does not directly reflect the soil water content, which is of crucial importance for plant development during the vegetation season. However, it is laborious and difficult to develop fine-resolution physiological variables for each particular study, and still in most cases the only easily accessible information sources are the classical meteorological data and derived bioclimatic variables [13].

The regression analysis revealed the significant effect of land use type on the plant diversity only in the Pazardzhik- Plovdiv region (*p* = 0.005, Table 2), while the same effect was not significant in the case of Western Stara Planina and on the pooled data. The effect of soil type was significant again only in the case of the Pazardzhik-Plovdiv region. 

The analysis of the effect of soil type was probably affected by the fact that many land use types (crops) were represented on a few soil types (i.e., the design did not allow proper statistical treatment). Each crop species has particular requirements regarding soil conditions and cannot be planted everywhere. Therefore, the evaluation of the effect of soil type on plant diversity in different land use types would require special experimental design [15], which was not available in the present study, because the study objective was different. The soil type definitely affects the plant diversity. However, in agricultural fields, soils are modified to a great extent by tillage, by applying fertilizers and by human activity in general. Testing the effect of soil type on plant diversity would be much more reliable if performed outside intensively managed agricultural fields.

The effect of land use type on plant diversity in the Pazardzhik-Plovdiv region, measured by the mean number of species per scoring plot, was tested by one-way ANOVA and was significant (*p* = 0.015) (Table 3). Even though such a test lowers the degrees of freedom significantly (only one mean value per land use type) and therefore is less reliable statistically, it still could be used as an additional criterion describing the effect of land use type on plant diversity. The highest diversity was recorded in group 7, which represents land with trees and shrubs, including forests. High diversity was found also in the wetlands (group 8) and in the horticultural fields. The lowest diversity was recorded in the cereal fields. When the species number was considered, the highest mean diversity was calculated for group 8 (wetlands), followed by the cereals and the wooded land; however, here the differences were not statistically significant (Table 3). 

When considering the separate land use types, the maximum total of 85 plant species were recorded in the scoring plots along the small to medium-sized streams and rivers (code E32), 69 species were found in the wheat fields (code A11), and 65 in the pastures (code C22); however, there were many more scoring plots in wheat, and therefore, the mean number of species per scoring plot was lower in comparison with the other two types (Table 2, Figure 1). Other land use types with relatively high plant diversity were those with sparse cover of trees and shrubs (D11 and E11), with 50 and 42 species, respectively. 

The highest diversity in Western Stara Planina was found in group 9 of land use types—roads and ditches, followed by group 6—meadows and hay fields. The other groups had similar values with statistically non-significant differences among means. When the mean number of species per scoring plot was considered, the results were more even, ranging from 12.7 (cereal fields) to 33.5 (wetlands). The mean number of species per scoring plot was higher again in group 9, followed by group 8 (wetlands). 

In the individual land use types, most species were recorded in C12 (Grazing land with sparse tree/shrub cover)—135, followed by E51 (dirt/gravel track)—92, and C11 (Meadow/hay field with sparse tree/shrub cover)—71 (Table 3, Figure 1). Considering also the high number of species recorded in the other similar land use types (shrubland, forest, alfalfa field), it was evident that all they were of semi-natural character, with the exception of the dirt road, which can be considered as an outlier. 

The ecosystems in Western Stara Planina can be described as less affected by human activities. The land use in Pazardzhik-Plovdiv region has a much longer history—most of the territory here has been extensively used as arable land for thousands of years, and the long period of human influence has affected the natural distribution of plant species. Pastures, meadows and wooded land, which can be considered as natural and semi-natural ecosystems, occupy a very small amount of that region, contrary to the Western Stara Planina region, where these land use types predominate (38.6% of the total). 

The effect of climate was better expressed than the effects of land use type and soil type. One bioclimatic variable had a significant effect on the plant diversity in the Pazardzhik-Plovdiv region, two variables in Western Stara Planina, and four bioclimatic variables affected significantly the plant diversity when the two regions were considered together. In the case of the Pazardzhik-Plovdiv region, the significant factor was the mean diurnal range, while in the Western Stara Planina region the most important climatic factors affecting the plant diversity were precipitation seasonality and precipitation during the warmest quarter, characterized with insufficient air and soil humidity. The tests on pooled data revealed four bioclimatic variables as having a significant effect on the plant diversity. Two of them were temperature-related (mean annual temperature and mean diurnal range), and two were related to precipitation distribution (precipitation of the wettest month and of the driest quarter). Evidently, pooling the climatic data of the two regions together allows better expression of trends due to the climate effect on the plant cover. Within a relatively small area, like the regions of this study, the differences in climatic variables are of small magnitude. An exception is a site in the Pazardzhik-Plovdiv region, near the village of Ravnogor, which is situated at 1200–1300 m a.s.l. and its climatic variables differ substantially from these of the other sites, situated in the plain at 200–300 m a.s.l. This site can be considered as an outlier, as the few land use types and soil types studied here could not contribute to a particular trend. 

When considering two separate regions representing different climate subtypes (continental in Western Stara Planina and transitional toward sub-Mediterranean in the Pazardzhik-Plovdiv region), the effect of climate variables on the plant diversity is expected to be much more pronounced. Also, the spatial scale is much wider and also encompasses various orographic characteristics related to diverse climate conditions. Therefore, detecting the climate effect at a wider scale is more realistic and sound.

The use of bioclimatic variables in plant species distribution modelling recently increased substantially [13]. Most studies, however, addressed broad-scale issues, related to the effect of global climate change [13,16,17,18,19]. At a local scale, the results of case studies could differ significantly, due to the variation in local environmental conditions. It is useful to combine the climatic variables with other factors, including anthropogenic ones. Land use tends to slow down the migration rate of plant species, due to the landscape fragmentation [20]. There are still a lot of uncertainties and limitations in bioclimatic modelling [21] that should be considered when applied to a particular case study [22,23].

The results revealed that the level of plant diversity depends on numerous factors and the land use type was not the most important. Overall, the relationships between land use and biodiversity in the broadest sense are very complex and highly context dependent [24]. 

The highest plant diversity, with 135 species, was recorded in the land use type C12 (grazing land with sparse tree cover) (Table 4, Figure 1). However, the mean species number per scoring plot was 7.1. Other land use types with high diversity were E51 (96 spp.), A62 (59 spp.), A21, E32 (see Table 4 for the names of land use types). Many of these territories are situated along rivers and small streams, often with tree vegetation. Analogously, the most important sites for the biodiversity conservation in an agricultural landscape in western Poland were the marshy and swamp meadows endangered by drying and intensification of agricultural activities [25]. This indicates the importance of Green and Blue infrastructures (GBI) as a part of agroecosystems. The results from this study are in line with those reported by Boutin et al. [26], who found greater diversity in the hedgerows. Walker et al. [27] outlined the high plant diversity of green lanes and concluded that it was affected by the environmental conditions, but also by the agricultural practices in the adjacent arable fields. The semi-natural boundaries of agricultural fields had a strong effect on the plant diversity in the Mediterranean cropland, too [28].

The lowest plant diversity in both regions was established in the land use types representing annual crops (corn, sunflower) or on fallow land. Intermediate values of plant species diversity were documented in the remaining land use types. These results are consistent with the generalization that the lowest ecosystem quality is detected in intensively used agricultural areas in lowlands [29]. 

The percentage of indigenous, alien and cultivated species in the two regions did not differ significantly. There were 93.2% native, 4.8% alien and 2% cultivated species in the Plovdiv-Pazardzhik region, while in Western Stara Planina these percentages were 94.5%, 1.9% and 3.6%, respectively. The ratio of indigenous species was still quite high, in comparison, for example, with a previous report from Poland [25].

The established database provides information about the most frequent weed species, which could be used in the planning of agricultural activities and weed management. The most common weeds in the Pazardzhik-Plovdiv region were *Chenopodium album*, *Cynodon dactylon*, *Setaria viridis* and *Echinochloa crus-galii*, found in more than 40% of the transects. Other weeds and ruderal species, represented by more than 20% of the transects, were *Convolvulus arvensis*, *Portulaca oleracaea*, *Taraxacum officinale*, *Xanthium strumarium*, *Berteroa incana* and *Galinsoga parviflora*. 

Contrary to the observed diversity in the Pazardzhik-Plovdiv region, the most frequent species in the Western Stara Planina region were not weeds but indigenous species such as *Agrimonia eupatoria*, *Achillea millefolium*, *Cichorium inthybus*, *Galium verum*, and *Daucus carota*. They were represented in more than 30% of the transects. Other frequently occurring species were *Dactylis glomerata*, *Rubus caesius*, *Crataegus monogyna*, *Mentha longifolia*, *Cirsium arvense*, *Prunus spinosa*, *Chondrilla juncea*, *Cirsium ligulare*, *Dipsacus laciniatus*, *Prunus cerasifera*, *Rubus idaeus*, *Convolvulus arvensis*, *Rosa canina*, recorded in more than 20% of the studied transects. 

An important characteristic of the two regions was the number of medicinal plants in the flora in both regions; 178 in the Pazardzhik-Plovdiv and 257 in the Western Stara Planina Mts. This fact suggests additional opportunities for use of the natural plant resources, besides the traditionally practiced agricultural activities. The results indicate that medicinal plants in the agricultural fields and their peripheral parts, represented relatively small percent of the species composition. The most frequent were 41 species found in 10% or more of the transects. Only about 22% of these species were medicinal plants. The highest diversity of medicinal plants was recorded in the remnants of woody vegetation (solitary trees and small groups of trees/bushes) and in fruit tree orchards, representing places that are not subjected to regular soil cultivation and/or herbicide application. However, the diversity of medicinal plant species in the Pazardzhik-Plovdiv region was relatively low. Besides the low level of species diversity in the agricultural land, pesticide use prevents the use of these species for medicinal purposes. 

The comparison between the two regions revealed that the indigenous flora of Western Stara Planina was more conserved, most probably due to the orographic characteristics of the region, which affect significantly the land use type. Eleven orchid species with conservation significance were found in the pastures and meadows of Western Stara Planina region: *Dactylorhiza sambucina*, *Gymnadenia conopsea*, *Ophrys cornuta*, *Orchis laxiflora*, *Orchis morio*, *Orchis pallens*, *Orchis papilionacea*, *Orchis purpurea*, *Orchis simia*, *Orchis ustulata*, and *Traunsteinera globosa*. These orchid species occurred in four land use types: open areas (meadows and grazing land) with sparse tree/shrub cover and in the forest (C11, C12, D11 and N11, Table 1). All these species are protected by CITES—seven of them are part of Appendix 3, and four species are part of Appendix 4 of [30]. One species, *Traunsteinera globosa*, is included in the Red Data Book of Bulgaria with the category ‘critically endangered’ [31]. No species of conservation importance were found in the experimental plots of Pazardzhik-Plovdiv region, which was most probably due to the extensive agricultural activities with the application of pesticides. 

The present study was the first assessment of the plant diversity across large-scale agroecosystems, representing various land use types in Bulgaria. The issues are gaining continuous importance and attention from scientists and policy makers [2,32,33]. However, in Bulgaria, and in the adjacent countries in the Balkans, biodiversity in agricultural land is still a neglected issue. Including this topic in the agenda of biodiversity studies could open up new opportunities for interdisciplinary research and could bring together researchers from the fields of biodiversity and agriculture. Most studies and reviews in the Balkans have focused on the high natural biodiversity, which is generally well-known [34]. However, the plant diversity in the agricultural lands in the region, and especially in relation to land use type, remains largely unknown. In this context, we consider the future cooperation among the countries in the region, each one with its own traditions in agriculture, as essential. Such investigations will contribute to policy making and to improving agricultural sustainability and biodiversity conservation. 

## 3. Materials and Methods 

### 3.1. Regions of Study

The two-year study was conducted in two regions in Bulgaria; in the Thracian plain between Plovdiv and Pazardzhik, and in the area of Western Stara Planina (the Balkan Mountains) and the adjacent territories. Selecting the two regions for the study was based on numerous criteria including agricultural and social issues, but also biodiversity and regional development [6,35]. 

The following land cover types were selected in the Corine land cover layer for Bulgaria; 211—arable land, 221—vineyards, 222—fruit trees, 231—pastures, 242—complex cultivation patterns, 243—land occupied by agriculture, with significant areas of natural vegetation. The Corine layer was clipped (Clip tool) for the case study areas (the municipalities in Western Stara Planina and Plovdiv region). The same procedure was repeated with the land use layers for the respective regions, after which both layers were merged (Merge tool). The case study areas were measured into a 3 km grid using the Create Fishnet tool with the optional Create Label Points box enabled. The fishnet label points were checked against the merged Corine/Landuse layer with the Intersect tool. The Create Random Points tool was used to determine the exact squares to be visited in the field for each region. The intersected fishnet label points were used as a constraining feature class, with the number of points set to 20 and minimum allowed distance set to 6 km. All procedures were done in ArcGis 10. The map of studied regions is presented in Figure 2.

Thirty-three land use types were investigated in the Pazardzhik-Plovdiv region and twenty-four in the Western Stara Planina. The predominant land use type in the Pazardzhik-Plovdiv region were cereal fields (23.6% of all scoring plots), followed by sunflower (9%). About 8% of the scoring plots were in pastures.

The field work was performed during the spring and autumn, in order to cover the whole aspect of the flora. A structural analysis of plant diversity was performed in experimental plots, representing more than 30 land use types. The sampling approach was based on the Landscape Infrastructure and Sustainable Agriculture (LISA) methodology [35]. Four transects were established in each experimental plot. Each transect was 30 m long and started at least 7 m within the boundaries of the respective land use type. The taxonomic treatments followed previous studies [36,37]. The codes of the respective land use types follow the ones specified in the LISA methodology (Table 1). Because some land use types predominate, there were different numbers of scoring plots per land use type. The number of species in a scoring plot was the main parameter recorded. Additional parameters were the number of species per land use type, and the mean number of species per scoring plot. The first and third parameters were used as measures of plant diversity. 

### 3.2. Effect of Soil Type on Plant Diversity 

Information about the soil types in the experimental plots in the different land use types was taken from the newly established database during the work on the project BIOGEA. Soil types follow the classification of FAO [38]. Also, additional studies were used for more details [39,40]. The following main soil types were identified in the two regions:

Pazardzhik-Plovdiv—Cambisols, Chromic cambisols, Eutric fluvisols, Eutric vertisols, and Gleyic vertisols;

Western Stara Planina—Cambisols, Albic luvisols, Haplic luvisols, Haplic luvisols – eroded, Haplic luvisols + Rendzic leptosols, Eutric fluvisols, Rendzic leptosols, and Gleyic luvisols.

### 3.3. Effect of Land Use Type on the Plant Diversity

The classification of land use types used in the field study is too detailed (Table 1). Therefore, for strengthening the analysis, the most similar land use types were pooled to form the following groups (codes in brackets follow these in Table 1): 1. Annual crops—cereals and sunflower (A11, A16, A31); 2. Annual crops—potatoes and tobacco (A21, A41); 3. Vegetables and wintering crops (strawberries) (A53, A54); 4. Arable land without crops—fallow land and unmanaged land (set aside) (B12 and A73); 5. Horticultural fields, rose strips (A74, A81, A82, A83, A84, A87, A92); 6. Meadows and hay fields (with or without sparse tree/shrub cover), incl. alfalfa (A62, C11, C12, C21, C22); 7. Trees and shrubs, incl. forests (C31.4, D11, E11, E12, E21, N11); 8. Wetlands (E31, E32, E33, E34); 9. Roads and ditches (E51, E71). The number of species and mean number of species per scoring plot were used as variables.

### 3.4. Effect of Climatic Conditions

Climatic conditions were included in the analyses in the form of 19 bioclimatic variables (Table 1) representing a broad range of parameters related to temperature and precipitation. The bioclimatic layers in GIS compatible format were downloaded from the Worldclim website [41].

### 3.5. Data Analysis

The first step was to shrink the number of bioclimatic variables (19) in order to make possible their analysis. For this purpose, we applied principal component analysis (PCA) using R statistical software. Ten selected bioclimatic variables were used for further analysis. Dependent variable (species diversity) was log-transformed before the analysis for improving the normality. Linear regression analysis was applied to detect the effect of the studied factors on plant diversity. The factors included in the model were: (1) land use type; (2) groups of land use types; (3) bioclimatic variables and (4) soil type. The analysis was performed on the data of each region separately, and on pooled data, because the climate differences within the relatively small area of the regions of study are less expressed than when both regions are considered.

One-way ANOVA was applied to test the effect of the grouping of land use types on the plant diversity, measured as mean number per scoring plot for a land use type, because these data could not be included in the analysis presented above. When the effect was proven to be significant (*p ≤ 0.05*), then post-hoc Fisher LSD test was applied to reveal the significant differences among means of different groups of land use types.

## Figures and Tables

**Figure 1 plants-09-00602-f001:**
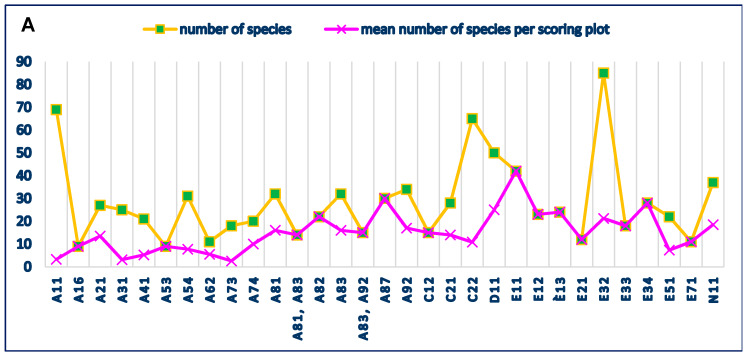
Plant diversity in the regions of study. (**A**)—Pazardzhik-Plovdiv Region; (**B**)—Western Stara Planina region.

**Figure 2 plants-09-00602-f002:**
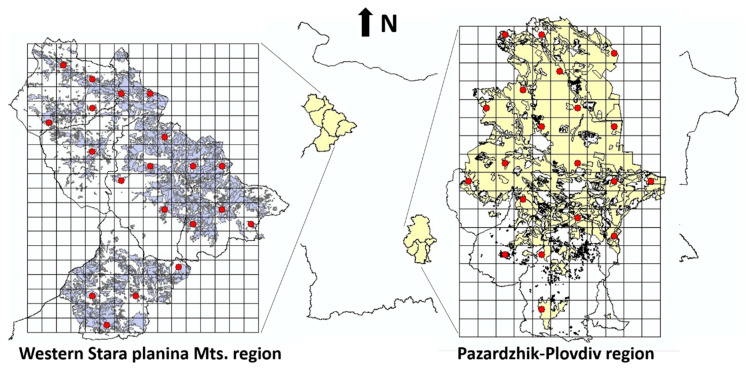
Maps of the two regions of study together with the land use types studied (red dots).

**Table 1 plants-09-00602-t001:** Summary of principal component analysis for the bioclimatic variables (pooled data for the two regions), including variable loadings for principal components 1–3.

Variable Name	PC1	PC2	PC3
1. Mean annual temperature	0.027	0.441	−0.003
2. Mean diurnal range	0.299	−0.031	−0.073
3. Isothermality	0.291	−0.009	−0.138
4. Temperature seasonality	0.015	−0.118	0.592
5. Max. temperature of the warmest month	0.047	0.349	0.324
6. Min. temperature of the coldest month	−0.086	0.404	0.141
7. Temperature annual range	0.210	−0.125	0.287
8. Mean temperature of the wettest quarter	−0.272	0.167	0.121
9. Mean temperature of driest quarter	0.176	0.285	−0.303
10. Mean temperature of warmest quarter	0.019	0.435	0.080
11. Mean temperature of coldest quarter	0.017	0.430	−0.092
12. Total (annual) precipitation	−0.309	−0.021	−0.033
13. Precipitation of wettest month	−0.297	−0.014	−0.158
14. Precipitation of driest month	−0.302	0.024	0.112
15. Precipitation seasonality	−0.131	0.008	−0.494
16. Precipitation of wettest quarter	−0.305	−0.025	−0.076
17. Precipitation of driest quarter	−0.307	−0.005	0.070
18. Precipitation of warmest quarter	−0.298	−0.064	−0.009
19. Precipitation of the coldest quarter	−0.308	0.013	0.054
Standard deviation	3.207	2.251	1.560
Proportion of variance	0.541	0.267	0.128
Cumulative proportion of variance	0.541	0.808	0.936

**Table 2 plants-09-00602-t002:** Significance of land use type and environmental variables on the plant diversity (*p*-values), as determined by linear regression analysis.

Factors	Plovdiv-Pazardzhik	Western Stara Planina	Pooled
Land use type	**0.005 ^b^**	0.520	0.080
Group of land use type	0.077	0.427	0.980
Soil type	**0.016**	0.369	0.437
Bioclim1	0.695	0.101	**0.047**
Bioclim2	**0.004**	0.694	**<0.001**
Bioclim5	0.900	0.105	0.077
Bioclim6	– ^a^	0.702	0.863
Bioclim7	0.203	0.334	0.117
Bioclim9	0.758	0.684	0.115
Bioclim10	0.152	0.880	0.840
Bioclim11	0.095	0.187	0.080
Bioclim13	– ^a^	0.177	**0.001**
Bioclim14	– ^a^	0.178	0.700
Bioclim15	0.422	**0.031**	0.751
Bioclim18	– ^a^	**0.039**	**0.015**

^a^ excluded from the analysis due to high correlation with other variables. ^b^ the significant effects (*p*-values ≤ 0.05) are presented in bold.

**Table 3 plants-09-00602-t003:** Effect of groups of land use types on the plant diversity in the two regions of study, when the mean number of species per scoring plot was considered as depending variable (tested by one-way ANOVA).

Pazardzhik-Plovdiv Region		Western Stara Planina Region	
GRL	Means	GRL	Means
1	8.7d ^1^	1	12.7b
2	15.4abcd	5	19.3ab
3	11.5cd	6	15.4b
5	20.2abc	7	14.1b
6	13.9bcd	8	26.5a
7	25.6a	9	33.5a
8	24.2ab		
9	11.5cd		
	F = 3.39; *p* = 0.015; d.f. = 7 **		F = 3.40; *p* = 0.04; d.f. = 5

GRL—group of land use types (for codes, see Material and Methods); ^1^ Means within a column followed by the same letter are not significantly different at *p* ≤ 0.05, as tested by one-way ANOVA, followed by Fisher LSD post-hoc test. ** The last row represents the results of one-way ANOVA.

**Table 4 plants-09-00602-t004:** Plant diversity by different land use types in the two regions of study.

Code According to LISA Methodology	Land Use Type	Number of Plant Species	
		Pazardzhik-Plovdiv	Western Stara Planina
**A11**	Wheat	69	32
A16	Maize	9	41
A21	Potatoes	27	29
A31	Sunflower	25	41
A53	Strawberries	9	-
A41	Tobacco	21	-
A54	Other fresh vegetables	31	46
A62	Alfalfa (Lucerne)	11	59
A73	Arable land without plants (e.g., recently sown)	18	-
A74	Flower areas and strips (*Rosa damascena* fields)	20	-
A81	Apple fruit	32	37
A82	Pear fruit	22	-
A83	Cherry fruit	32	-
A84	Nuts trees	- *	22
A87	Other fruit trees and berries	30	-
A92	Vineyards	34	15
B12	Unmanaged set-aside		21
C11	Meadow/hay field with sparse tree/shrub cover	-	71
C12	Grazing land with sparse tree/shrub cover	15	135
C21	Meadow/hay field without sparse tree/shrub cover	28	-
C22	Grazing land without sparse tree/shrub cover	65	-
C31.4	Meadow orchard stand with greater gaps (cov. 50–75%)	-	32
D11	Shrubland with sparse tree cover	50	57
E11	Solitary trees and small groups of trees/bushes	42	-
E12	Tree lines and avenues	23	18
E13	Hedges and bushes	24	-
E21	Buffer strips	12	2
E31	Springs and spring swamps	-	21
E32	Small and medium-sized flowing waters (streams, rivers)	85	32
E33	Ditches (flowing and standing water)	18	-
E34	Small water bodies (Ponds, ponded depressions, and pools)	28	-
E51	Dirt/gravel track	22	92
E71	Ditches	11	-
N11	Forest	37	45

* The respective land use type was not found in the experimental plot areas.

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
