# Peer review of "Influence of the Land Use Type on the Wild Plant Diversity"

_plants, 2020, doi:10.3390/plants9050602_

Round 1
Reviewer 1 Report
I see the revised manuscript “Influence of the land use type on the wild plant diversity” resubmitted for pubblication in Plants Journal. The authors responded to the reviewer's major comment. I am satisfied with the manuscript revised in the present form.
Author Response
We would like to express our thanks to Reviewer 1 for his positive evaluation of our paper!
Reviewer 2 Report
I have read the resubmitted version of the manuscript from Aneva et al. with interest. Although the manuscript was improved by the Authors, I still think it lacks the right balance between being an ecologically or botanically oriented. Clearly, the Authors tried to drive the balance more on ecology, with a bit of statistical analysis. I appreciate the effort, but I think the results still need a vast improvement. The (very) simple ANOVA approach by the Authors is clearly not adequate to dealing with this approach. I suggest the Authors to try a GLMM approach, using not only land use and/or soil type as factors, but also rely on bioclimatic variables, which the Authors can gain for free at worldclim.org. As a matter of fact, I expect that, beside land use and soil type, climate is the driving variable in their observed pattern of biodiversity. I also encourage the Authors to improve their tables (no measure of variability is reported) as well as their figures. I find that Figures 1&2 arevery low quality and not very informative. Once the Authors truly find a way to make this manuscript appealing to an international audience by means of a testable hyphothesis, then I think it might be considered for publication.
Author Response
We would like to express our thanks to the reviewers. We provide the following answers and comments to the remarks and recommendations:
Remark: The research design must be improved.
Answer: We did substantial changes and improvement in the research design.
First, we got (downloaded) 19 Bioclimatic variables from Worldclim.org. They concern the experimental plots where the field studies took place. Therefore, in this case we considered each land use type within the context of a particular experimental plot, situated on a particular soil and under particular climate conditions. This is radically different from the previous variant, where for each land use type we used pooled data from all experimental plots within a region.
Second, the variable “mean species number per sample plot” could not be used in this context, and in order not to leave some information unused, we kept the results using this indicator (only one-way ANOVA was possible)
Third, we did analyses for each region separately and on data pooled for the two regions. Thus, the effect of climate conditions became more pronounced. This was necessary because the regions are not big and the existing differences in bioclimatic variables are sometimes (or often) of a small magnitude that is difficult to be caught by the statistical treatment.
Remark: Research methods description could be improved.
Answer: We tried to improve the description of the research methods by reporting the main steps of the analysis. The first step was to shrink (reduce) the number of bioclimatic variables (19) in order to make possible their analysis. For this purpose, we applied Principal Component Analysis (PCA) using R statistical software. Based on the results we applied further linear regression analysis using log-transformed data of the depending variable (species diversity), in order to improve the normality. Applying of GLMM seemed problematic because immediate difficulties arise in determining the random factor(s). The factors used in the model were: 1) Land use type (LUT); 2) LUT group; 3) Bioclimatic variables and 4) Soil type. According to our limited knowledge of statistics, these seem fixed factors. We tried to provide more details concerning the research methods in the end of the paper.
Remark: Presentation of the results must be improved.
Answer: We added the results of the effect of bioclimatic variables and discussed them in the context of the international studies.
Remark: Are the conclusions supported by the results? Can be improved.
Answer: Although the conclusions are not outlined as a separate part, our attempt was to provide conclusions based solely on the results of the present study. We extended the modest conclusions of the previous variant by adding few sentences about the effect of bioclimatic variables. Accordingly, the Abstract was modified in order to encompass all the corrections and, hopefully, improvements done in the papers.
The figures 1 and 2 were simplified and pooled into one figure with two parts.
And finally, if necessary, we could change the title to: “Effect of environmental factors and land use type on the wild plant diversity” in order to correspond better to the content of the paper.
This manuscript is a resubmission of an earlier submission. The following is a list of the peer review reports and author responses from that submission.
Round 1
Reviewer 1 Report
Brief summary
The study shows the wild plant diversity across the agroecosystems in two contrasting regions of Bulgaria; Pazardzhik-Plovdiv (representing agroecosystems in the lowlands) and Western Stara Planina (the Balkan Mountains, representing agroecosystems in the foothills of the mountains). The results of plant diversity were related on the land use type. The highest plant diversity was found on grazing land with sparse tree cover, while the lowest one – in the land use types representing annual crops or fallow. The study also established a database on weed species, indigenoeus species and medicinal species from the two areas study. Based on comparation data, the study highlights that the indigenous flora of Western Stara Planina was more conserved; eleven orchid species with conservation significance were found in the pastures and meadows in that region. The study is the first attempt in Bulgaria to characterize the plant diversity across diverse agroecosystems representing many different land use types.
Broad comments
1)The article is original and quite well written in every section. It represents an important contribution to the wild diversity of agricolture lands in Bulgaria where this topic was neglected. There are many pubblished studies in Europe on plant diversity conducted in natural and semi-natural environments but relatively few on agriculture lands based on land use type. These studies are important especially if we consider that agriculture has a strong impact on habitats and landscape biodiversity.
2)The article is original even if it is known that the lowlands are generally more anthropized and consequently have less floristic diversity than the sectors characterized by a higher morphological diversity and less anthropic impact. This is evidenced by the high number of indigenous species including numerous orchids that distinguishes Western Stara Planina (the Balkan Mountains, representing agroecosystems in the foothills of the mountains) compared to Pazardzhik-Plovdiv (representing agroecosystems in the lowlands).
3) In the Material and Methods section (or in Supplementary material) I suggest to add a summary table of the environmental characteristics (climate, altitude, substrata, etc) of the two regions (Pazardzhik-Plovdiv and Western Stara Planina) including also the density of the resident population. This would be useful to better understand the results of the biodiversity comparison between the two areas under study
4)In the abstract (line 21) and in the introduction (line 64), the authors say biodiversity depends on numerous factors but that the most important is the type of land use. I suggest to emphasize in the summary also the environmental factor in the comparison between the two areas as regards the conservation of indigenous species
Specific comments:
Line 61 - Quote [7] is not Gussav et al. but it is Duel and Obrist. Please correct.
Reviewer 2 Report
The ms. “Influence of the land use type on the wild plant diversity (Manuscript ID: plants-692334)”, by Ina Aneva et al. evaluates the wild plant diversity across the agroecosystems in two contrasting regions of Bulgaria. Although an interesting topic, the manuscript lacks a clear objective of the study performed, namely: Why the use of such regions? For what purpose?. Also the authors have generated a set of data that is of interest, but they should rethink the goals of their study and perform additional analyses in order to produce a manuscript that is of more scientific interest. As it is, the paper is merely descriptive and only have a regional interest related to the land use types in Bulgaria.
Reviewer 3 Report
I have read with interest the manuscript from Aneva et al.
I have to say that, being an ecologist with a strong background in botany, this study appeared to me as a bit pointless. As an ecologist, I cannot find many uses for this study, because there is a complete lack of statistics that might help to prove any hypothesis. As it is, even the title is wrong in this context, as the Authors have no way to assert the "influence of the land use type on the wild plant diversity", because they do not use any statistics that might help us if the effects are given to climate, soil type and so on. On such a large scale and without a hypothesis, the Authors just present a sort of catalogue. And then, as a botanist, I cannot find any use for such catalogue, as this is neither a floristic or a vegetation study, and the results are presented rather blandly. I suggest to rethink the whole research completely, as the data are a lot and I am sure the Authors can come up with a better way to present their results to the scientific community, but with a clear ecological or botanical idea in mind and with the support of a statistics colleague.